# A Study on the Fused Deposition Modeling Process of Graphene/Nano-Fe$_3$O$_4$ Composite Absorber and its Absorbing Properties of Electromagnetic Microwave

**Haihua Wu [1,*], Lei Xing [1,*], Yu Cai [1], Li Liu [1], Enyi He [1], Bo Li [1] and Xiaoyong Tian [2]**

[1] College of Mechanical and Power Engineering, China Three Gorges University, Yichang 443002, China; cy94824398@163.com (Y.C.); song295021725@gmail.com (L.L.); heenyi@ctgu.edu.cn (E.H.); liboiec@163.com (B.L.)

[2] State Key Laboratory for Manufacturing Systems Engineering, Xi'an Jiaotong University, Xi'an 710054, China; leoxyt@mail.xjtu.edu.cn

[*] Correspondence: wuhaihua@ctgu.edu.cn (H.W.); xing435324368@163.com (L.X.)

**Abstract:** Graphene/polylactic acid; nano-Fe$_3$O$_4$/polylactic acid; and graphene/nano-Fe$_3$O$_4$/polylactic acid composite absorbers are independently produced by fused deposition modeling technology. The effects of the content of graphene and nano-Fe$_3$O$_4$ on absorbing properties are investigated. After measuring the electromagnetic parameters using the waveguide method, the absorbing property is characterized according to the transmission line theory. The distribution of graphene and nano-Fe$_3$O$_4$ in the matrix is observed by scanning electronic microscopy (SEM). The results show that the graphene and nanometer ferroferric oxide multicomponent absorbing agent helps to form a synergistic absorbing effect. In the frequency range 8.2–18.0 GHz; the absorber has the greatest absorbing property when the content of graphene and nanosize Fe$_3$O$_4$ are 5 wt% and 20 wt%, respectively.

**Keywords:** fused deposition modeling process; graphene; nano-Fe$_3$O$_4$; absorbing properties

## 1. Introduction

Absorbing material is a functional material that converts electromagnetic waves into thermal energy or other forms of energy and has small reflection and transmission [1]. Generally, absorbing material is composed of an absorbing agent and substrate; [2,3]. According to the loss mechanism, the absorbing agent is divided into the following three types: First, the dielectric-type absorbing agent, such as FeN and Si$_3$N$_4$, which achieves the absorption effect through dielectric polarization relaxation loss; second, the magnetic-type absorbing agent such as ferrites, which attenuates electromagnetic waves through magnetic loss; and third, the resistive-type absorbing agent that absorbs electromagnetic waves by the attenuation of the electronic polarization or interfacial polarization of the medium such as carbon materials. The substrate is divided into metal (Al and Cu), non-metal (ceramic and carbon substrate), and polymer (thermoplastic and thermosetting).

Zhang et al. used graphene as the absorbing agent and polymethyl methacrylate as the matrix, and the graphene/polymethyl methacrylate absorber was produced by melt blending. The results showed that when the volume fraction of graphene was 1.8 vol%, the graphene formed a conductive network in the polymethyl methacrylate matrix, thus, the shielding efficiency of electromagnetic interference was 13 to 19 dB, in the frequency range 8–12 GHz [4]. Zheng et al. prepared a super paramagnetic ferroferric oxide (Fe$_3$O$_4$) nanocrystal using the in situ thermal decomposition method, and then mixed it with a paraffin matrix to prepare a Fe$_3$O$_4$/paraffin composite absorber (the mass fraction of Fe$_3$O$_4$ nanocrystal was 50 wt%). The test results showed that, when the thickness was 1 to

5 mm, the minimum reflection loss value was −4.4 dB, in the range 2–18 GHz. As many atoms exist on the surface of $Fe_3O_4$ nanocrystals, local symmetry is damaged. Thus, the polarization of dipoles was increased and $Fe_3O_4$ nanocrystals had super paramagnetic property which was beneficial for absorbing the electromagnetic waves [5]. He et al. prepared a graphene/$Fe_3O_4$ composite absorber using a solvothermal synthesis method (the ratio of graphene to $Fe_3O_4$ was 1:10). The test results showed that the reflection loss value of the graphene/$Fe_3O_4$ composite absorber with a thickness of 2 mm was less than −10 dB, and the minimum reflection loss value was −15.38 dB, in the frequency range 10.4–13.2 GHz [6]. The aforementioned researches covered numerous efforts to improve wave absorbing performance through the existence of synergy between graphene and $Fe_3O_4$.

Currently, the manufacturing methods of the absorber mainly include an etching method and a machining method. The former has higher costs and has a limit on the size of the absorber; the latter is difficult to meet the requirements of the complex structure of the absorber, and the processing cycle is long, and sometimes special molds or tools are required. Three-dimensional (3D) printing technology is an advanced manufacturing technology which builds objects by modeling layered processing of stacked materials. This technology can realize the rapid manufacturing of complex products with any structure, as well as the combination of multiple materials and structures and is expected to provide a new process technology for efficient microwave absorbers. In this paper, graphene/nano-$Fe_3O_4$/polylactic acid (PLA), nano-$Fe_3O_4$/PLA, and graphene/PLA composite absorbers are made by fused deposition modeling (FDM) technology. Electromagnetic parameters of the FDM samples are measured by the waveguide method, and the absorbing property is calculated according to the transmission line theory. Then, wave absorbing mechanism of the graphene/nano-$Fe_3O_4$ absorber is revealed based on analyzing the effects of different material compositions and content on the absorbing property

## 2. Experiments

### 2.1. Sample Preparation

If the graphene or nano-$Fe_3O_4$ content is low, it is difficult to reflect its absorption characteristics. When the content is too high, 3D printed composite wires cannot be formed. Therefore, we selected 3D printed composite wire of graphene/nano-$Fe_3O_4$/PLA (graphene content of 5 wt% and nano-$Fe_3O_4$ content of 20 wt%), nano-$Fe_3O_4$/PLA (nano-$Fe_3O_4$ content of 10 wt%, 20 wt%, and 30 wt%, respectively), and graphene/PLA (graphene content of 5 wt%, 7 wt%, and 9 wt%, respectively) with a diameter of 1.75 ± 0.05 mm, respectively [7]. Then, according to the requirements of test size, the composite absorbers with the size of 22.90 × 10.20 × 3.00 mm (X-band 8.2 to 12.4 GHz) and 15.90 × 8.03 × 3.00 mm (KU band 12.4 to 18.0 GHz) are produced by using the composite wire mentioned above. To ensure the print quality, the process was carried out with the following parameters: printing temperature is 160 °C, the printing speed is 40 mm/s, the printing filling density is 100%, the printing filling structure is straight, the ambient temperature is 24 °C, and the printing layer height is 0.1 mm.

### 2.2. Characterization and Testing

The cross-sectional morphology of the composite absorber was observed using a JSM-7500F field emission scanning electron microscope (Japan Electronics). The conductivity of the composite absorber was tested with a TH2512B DC low resistance tester (Beijing Crown Test Equipment Co., Ltd. Yongfeng Information Industry Base, Haidian District, Beijing, China). The electromagnetic parameters of the composite absorber in the frequency range of 8.2 to 18.0 GHz were tested by the waveguide method. The test instrument was a test system which was composed of a HP8720ES vector network analyzer (Zhongshan Huayitong Electronic Instrument Co., Ltd. Yihu Peninsula, Zhongshan, China) and corresponding waveguide. According to the measured electromagnetic parameters, the transmission line theory was used to calculate and analyze their reflection loss. The formulas are as follows [8]:

$$Z_0 = \sqrt{\mu_0 / \varepsilon_0} \tag{1}$$

$$Z_{in} = Z_0 \sqrt{\mu_r / \varepsilon_r} tanh\left(j2\pi f d \sqrt{\mu_r \varepsilon_r} / c\right) \tag{2}$$

$$RL = 20log\left|(Z_{in} - Z_0) / (Z_{in} + Z_0)\right| \tag{3}$$

where $\mu_0$ stands for complex permeability of free space, $\varepsilon_0$ is complex permittivity of free space, $Z_0$ represents input impedance of free space, $\mu_r$ is complex permeability of material, $\varepsilon_r$ is complex permittivity of material, $Z_{in}$ stands for normalized input impedance of material, $d$ is material thickness, $c$ represents speed of light in vacuum, and *RL* is material reflection loss.

## 3. Results and Discussion

### 3.1. Effect of Graphene Content on the Property of Graphene/PLA Composite Absorbers

The reflection loss of graphene/PLA composite absorbers with different graphene content is presented in Figure 1. It can be seen that, in the frequency range 10.0–15.8 GHz, the reflection loss of 7 wt% graphene/PLA composite absorber is lower than −4 dB and the minimum reflection loss value is −8.3 dB at the frequency of 13.2 GHz; in the frequency range 10.2–15.2 GHz, 9 wt% graphene/PLA composite absorber has a reflection loss of less than −4 dB and a minimum reflection loss of −7.3 dB at the frequency of 12.2 GHz. The reflection loss of 5 wt% graphene/PLA composite absorber is higher than −4 dB in the whole frequency band tested, reaching the minimum reflection loss value of −3.1 dB at the frequency of 14.2 GHz.

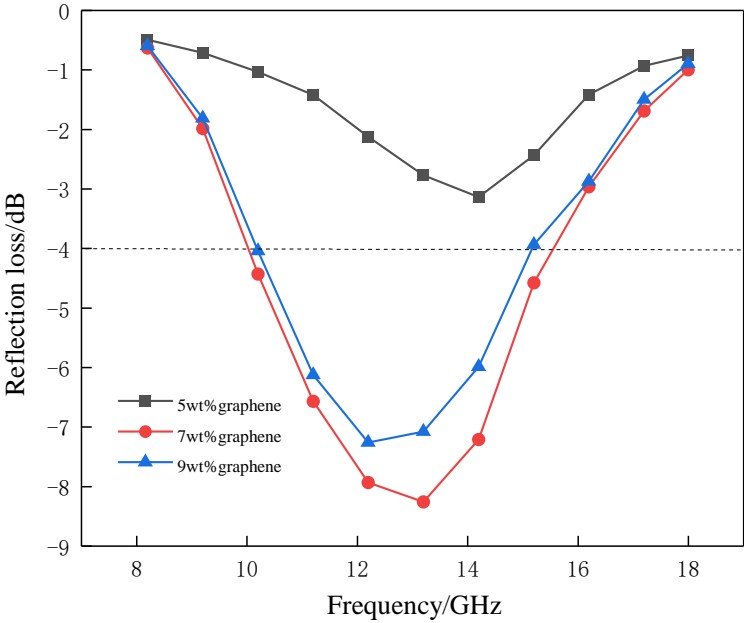

**Figure 1.** Reflection loss of graphene/PLA composite absorbers with different graphene content.

As shown in Figure 2, the real and imaginary parts of the complex permittivity of graphene/PLA composite absorber decrease with an increase of frequency. In Figure 2a, as the mass fraction of graphene increases, the real part of the complex permittivity of the composite absorber increases. When the graphene content is 9 wt%, the real part of the complex permittivity is the highest, and at 5 wt% it is the lowest, because as the mass fraction of graphene increases, the number of electric dipoles stored in the composite absorber increases, and the polarization of the electric dipole increases, resulting in an increase in the real part of the complex permittivity [9,10]. As shown in Figure 2b, when the graphene content is 7 wt%, the composite absorber has the highest complex permittivity imaginary

part, followed by 9 wt% and the lowest which is 5 wt%. According to the theory of free electrons [11], $\varepsilon'' = \sigma/\omega\varepsilon_0$, where $\sigma$ is the electrical conductivity of the material, $\omega$ is the pi, and $\varepsilon_0$ is the complex permittivity of free space. It can be concluded that the imaginary part of the complex permittivity is positively correlated with the conductivity. The electrical conductivity of the composite is $2.06 \times 10^{-4}$ S/cm when the content of graphene is 5 wt%. When the amount of graphene added is 7 wt%, the electrical conductivity is significantly increased to $2.71 \times 10^{-3}$ S/cm, indicating that the density of the three-dimensional conductive network is increased when the graphene content is increased from 5 wt% to 7 wt%. When the graphene content is 9 wt%, the electrical conductivity of the composite is $2.59 \times 10^{-3}$ S/cm. Due to the enhanced adsorption between graphene, the dispersion state of graphene in the matrix is affected and the conductive network structure is destroyed, and thus the conductivity is dropped.

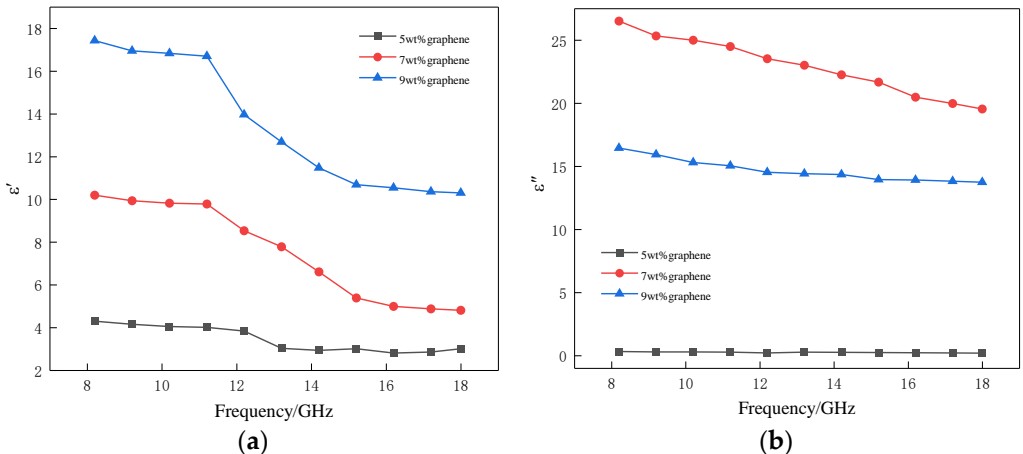

**Figure 2.** Complex permittivity of graphene/PLA composite absorbers with different graphene content. (**a**) The real part of the complex permittivity; (**b**) the imaginary part of the complex permittivity.

The SEM sectional view of PLA absorber and graphene/PLA composite absorbers are shown in Figure 3. It can be seen from Figure 3a that the pure PLA absorber has a relatively flat section. It can be seen from Figure 3b that the graphene is coated on the surface of the PLA matrix. Due to the stacking phenomenon between the graphene sheets, the wrinkles are more obvious when the amount of graphene is 7 wt% (as shown in Figure 3c). When the amount of graphene added is 9 wt%, as shown in Figure 3d, the graphene powders are agglomerated together and the absorbing property of the composite is weakened due to the presence of electron scattering. It can be seen that in the 8.2 to 18.0 GHz band, when the thickness of the composite part is constant, the absorbing property of the 7 wt% graphene/PLA composite is relatively better.

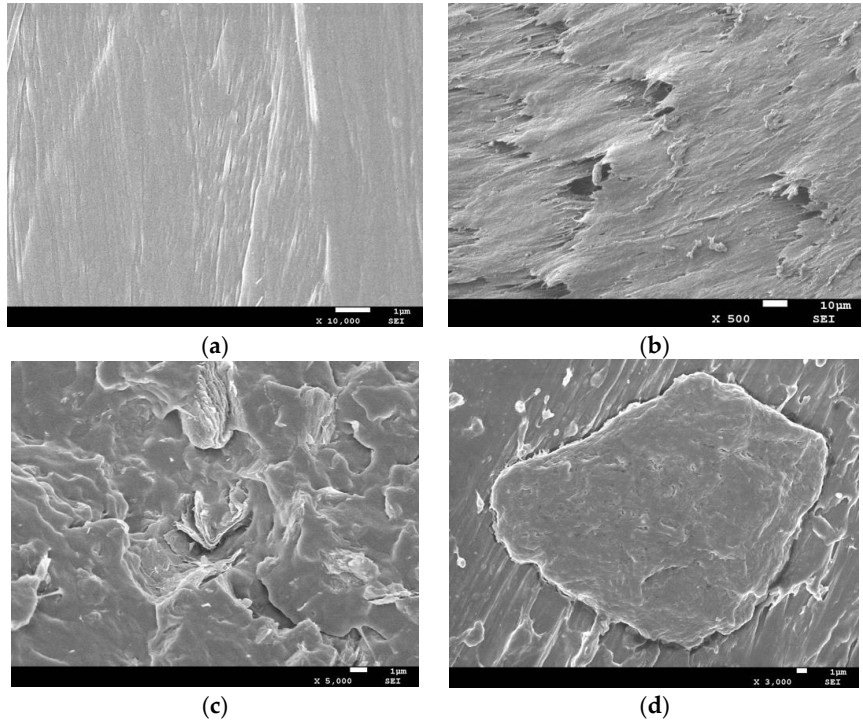

**Figure 3.** SEM cross section of pure PLA absorber and graphene/PLA composite absorbers. (**a**) Pure PLA absorber; (**b**) 5 wt% graphene/PLA composite absorber; (**c**) 7 wt% graphene/PLA composite absorber; and (**d**) 9 wt% graphene/PLA composite absorber.

*3.2. Effect of Nano-Fe$_3$O$_4$ Content on the Property of Nano-Fe$_3$O$_4$/PLA Composite Absorbers*

Comparisons of the reflection loss of the composite absorbers are shown as Figures 1 and 4, respectively. The variation is similar to that of graphene/PLA composite absorbers. The difference is that the minimum reflection loss value is shifted to the high frequency band. In the band of 8.2 to 18.0 GHz, when the content of nano-Fe$_3$O$_4$ in the nano-Fe$_3$O$_4$/PLA composite absorber is 20 wt%, the reflection loss curve is the lowest, followed by 30 wt%, and finally 10 wt%. The minimum reflection loss values achieved −4.8 dB, −3.9 dB, −3.3 dB, respectively.

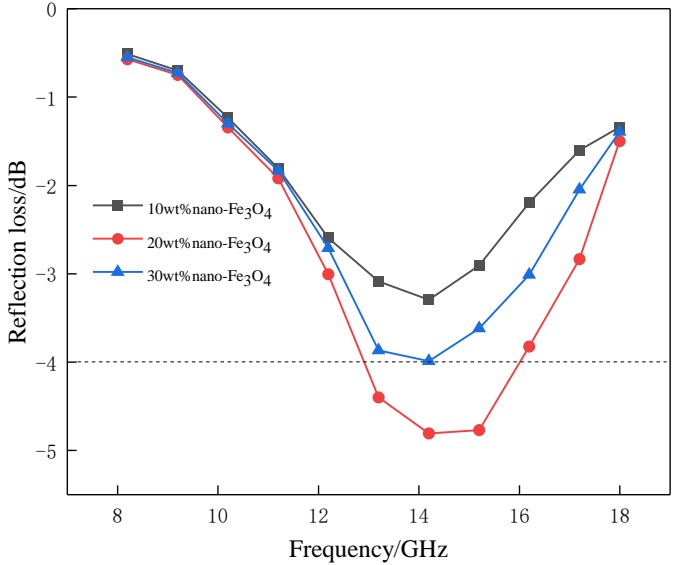

**Figure 4.** Reflection loss of nano-Fe$_3$O$_4$/PLA composite absorbers with different nano-Fe$_3$O$_4$ content.

As shown in Figure 5a,b, the complex permittivity of the nano-Fe$_3$O$_4$/PLA composite absorber decreases with increasing frequency. The complex permittivity can be expressed as $\varepsilon = \varepsilon_\infty + (\varepsilon_0 - \varepsilon_\infty)/(1 + i\omega\tau)$, where $\varepsilon_0$ and $\varepsilon_\infty$ are the complex dielectric constants of $\omega \to 0$ and $\omega \to \infty$, respectively, $\tau$ is the relaxation time, and $\omega$ is the circular frequency [12–15]. It can be seen that the complex permittivity $\varepsilon$ decreases as the frequency $f$ (relative to the circular frequency $\omega$) increases. In Figure 5a, with an increase of the mass fraction of nano-Fe$_3$O$_4$, the real part of the complex permittivity is improved. When the content of nano-Fe$_3$O$_4$ is 30 wt%, the complex permittivity is the highest, it is the second highest when the content is 20 wt%, and it is the lowest when the content is 10 wt%. This is explained by the fact that as the mass fraction of nano-Fe$_3$O$_4$ increases, the number of multi-interfaces formed by nano-Fe$_3$O$_4$ adsorbed on the surface of PLA and the interface polarization are increased, which leads to the increase in the real part of the complex permittivity. As shown in Figure 5b, when the content of nano-Fe$_3$O$_4$ in the nano-Fe$_3$O$_4$/PLA composite absorber is 20 wt%, the complex permittivity imaginary part is the highest, followed by when the content is 30 wt%, and it is the lowest when the content is 10 wt%. The reason for this is that as the amount of nano-Fe$_3$O$_4$ increased, the number of dipoles in the composite absorber increased, which caused the dipole polarization effect of the material when the addition amount of nano-Fe$_3$O$_4$ reached 30 wt%. Due to the enhanced adsorption between nano-Fe$_3$O$_4$, the dispersion state of nano-Fe$_3$O$_4$ in the matrix is affected, which weakens the ability of the dipole to lose electromagnetic waves.

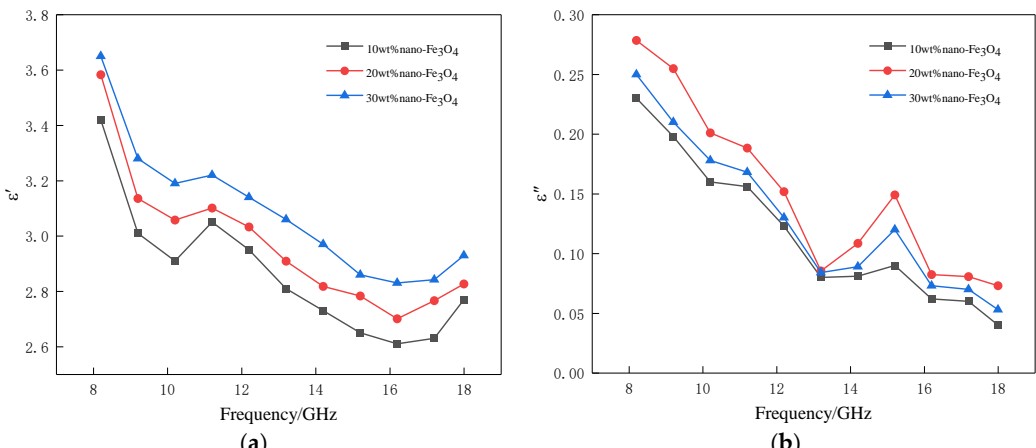

**Figure 5.** Complex permittivity of nano-Fe$_3$O$_4$/PLA composite absorbers with different nano-Fe$_3$O$_4$ content. (**a**) The real part of the complex permittivity; (**b**) the imaginary part of the complex permittivity.

Figure 6 is a graph showing the complex permeability of nano-Fe$_3$O$_4$/PLA composite absorbers with different nano-Fe$_3$O$_4$ content. It can be seen from Figure 6a that with an increase of frequency, the real part of the complex permeability gradually decreases, because the speed of the dielectric magnetization in the nano-Fe$_3$O$_4$/PLA composite absorber cannot keep up with the speed of the magnetic field change, forming a hysteresis phenomenon [16]. When the content of nano-Fe$_3$O$_4$ in the nano-Fe$_3$O$_4$/PLA composite absorber is 30 wt%, the real part of the complex permeability is the highest, followed by when the content is 20 wt%, and the lowest is when the content is 10 wt%. The increase of nano-Fe$_3$O$_4$ brings about improvement in magnetic storage. Therefore, the greater the content of nano-Fe$_3$O$_4$, the higher the value of the complex permeability real part. It can be seen from Figure 6b that the imaginary part of the complex permeability increases first, and then decreases; there exists a drastic change at a frequency of 16 GHz, because nano-Fe$_3$O$_4$ produces the ferromagnetic resonance phenomenon at higher frequencies. When the content of nano-Fe$_3$O$_4$ in nano-Fe$_3$O$_4$/PLA composite absorber is 20 wt%, the imaginary part of complex permeability is the highest, followed by when the content is 30 wt%, and the lowest when the content is 10 wt%. When the content of nano-Fe$_3$O$_4$ in the nano-Fe$_3$O$_4$/PLA composite absorber is 10 wt%, the polarization of Fe$^{2+}$ ions in nano-Fe$_3$O$_4$ is weak. When the addition amount of nano-Fe$_3$O$_4$ is 20 wt%, the polarization of Fe$^{2+}$ ions in nano-Fe$_3$O$_4$ is

enhanced, which causes the improvement of magnetic loss. When the content of nano-Fe$_3$O$_4$ is 30 wt%, a large amount of nano-Fe$_3$O$_4$ is agglomerated together, the magnetic loss of the electromagnetic wave is weakened, and therefore the imaginary part of the complex permeability has dropped.

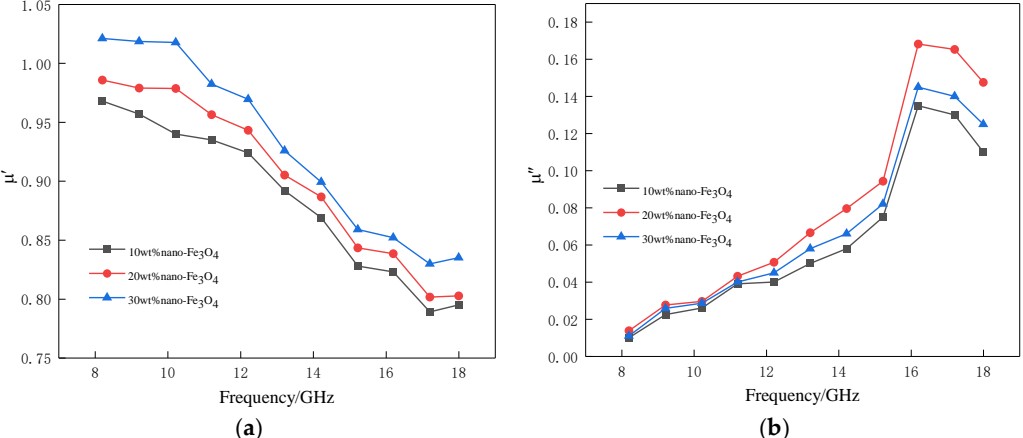

**Figure 6.** Complex permeability of nano-Fe$_3$O$_4$/PLA composite absorbers with different nano-Fe$_3$O$_4$ content. (**a**) The real part of the complex permeability; (**b**) the imaginary part of complex permeability.

One of the earliest ferrites that people have come into contact with is Fe$_3$O$_4$. It has high dielectric and magnetic properties, etc., widely used in the field of absorbing waves [17]. Figure 7 shows a SEM cross-section of pure PLA and nano-Fe$_3$O$_4$/PLA composite absorbers. As can be seen from Figure 7a, the pure PLA section has no holes. When the content of nano-Fe$_3$O$_4$ is 10 wt%, as shown in Figure 7b, the nano-Fe$_3$O$_4$ magnetic microspheres begin to adsorb on the surface of PLA, forming "protrusions" with pores. These "protrusions" promote multiple reflection losses of electromagnetic waves inside the material, while holes can act as polarization centers, generating polarization and relaxation under varying electromagnetic fields, thereby consuming electromagnetic energy. When the addition amount of nano-Fe$_3$O$_4$ is 20 wt%, as shown in Figure 7c, more nano-Fe$_3$O$_4$ magnetic microspheres are adsorbed on the PLA surface, which promotes the loss and absorption of electromagnetic waves. When the amount of nano-Fe$_3$O$_4$ added reaches 30 wt%, as shown in Figure 7d, the nano-Fe$_3$O$_4$ magnetic microspheres are agglomerated and the pores become larger. Agglomeration affects the ability of electromagnetic wave loss, since large holes allow electromagnetic waves to pass through the material directly, which weakens the absorption performance.

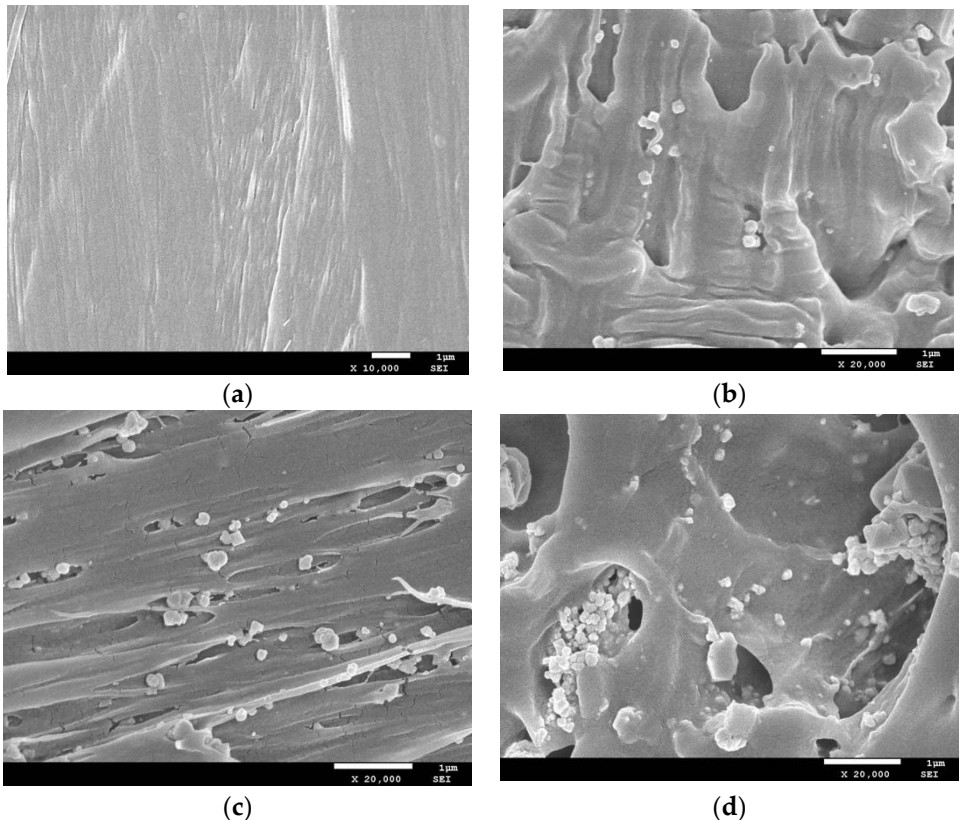

**Figure 7.** SEM cross-section of pure PLA and nano-Fe$_3$O$_4$/PLA composite absorbers. (**a**) Pure PLA absorber; (**b**) 10 wt% nano-Fe$_3$O$_4$/PLA composite absorber; (**c**) 20 wt% nano-Fe$_3$O$_4$/PLA composite absorber; and (**d**) 30 wt% nano-Fe$_3$O$_4$/PLA composite absorber.

### 3.3. Effect of Absorbing Agent Composition on the Property of Composite Absorbers

The reflection loss of composite absorbers with different absorbing agent composition is presented in Figure 8. It can be seen that as the frequency increases, the variation of the reflection loss of the three composite absorbers first decreases, and then rises. Among them, the reflection loss of graphene/nano-Fe$_3$O$_4$/PLA composite absorber is significantly lower than that of graphene/PLA and nano-Fe$_3$O$_4$/PLA composite absorbers, and the effective absorption bandwidth (RL ≤ −10 dB) of the graphene/nano-Fe$_3$O$_4$/PLA composite absorber is 3.6 GHz (12.4 to 16.0 GHz).

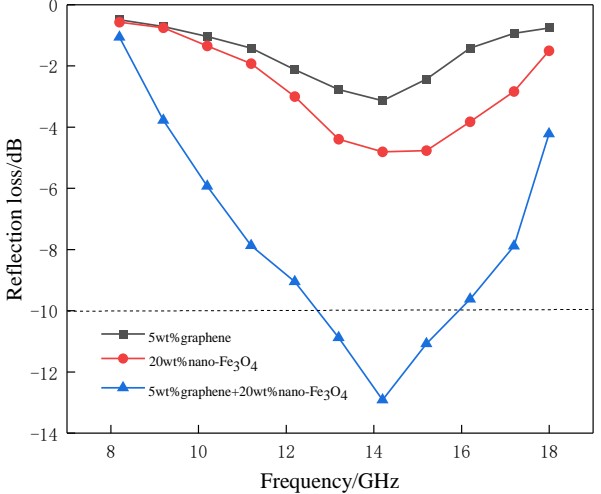

**Figure 8.** Reflection loss of composite absorbers with different absorbing agent composition.

The complex permittivity of composite absorbers with different absorbing agent composition are presented in Figure 9. Their variation law is gradually decreasing with increasing frequency, which is due to the relaxation phenomenon of the dielectric polarization in the composite absorber which cannot keep up with the changing speed of the electric field [18]. It can be seen from Figure 9a that the real part of the complex permittivity of the graphene/nano-$Fe_3O_4$/PLA composite absorber is higher than that of the graphene/PLA and nano-$Fe_3O_4$/PLA composite absorbers, because the real part of the complex permittivity of the material mainly depends on the interfacial polarization between the absorbing agent and the substrate. The graphene/nano-$Fe_3O_4$/PLA composite absorber increases the graphene/PLA interface and the graphene/nano-$Fe_3O_4$ interface with respect to the nano-$Fe_3O_4$/PLA composite absorber; the graphene/nano-$Fe_3O_4$/PLA composite absorber increases the nano-$Fe_3O_4$/PLA interface and the graphene/nano-$Fe_3O_4$ interface relative to graphite/PLA composite absorber. The increase of the interface leads to the highest real part of the complex permittivity. Secondly, since the dielectric property of graphene is significantly better than those of nano-$Fe_3O_4$, the polarization of graphene/PLA interface is stronger than that of nano-$Fe_3O_4$/PLA interface. Therefore, the real part of the complex permittivity of graphene/PLA composite absorber is higher than that of nano-$Fe_3O_4$/PLA composite absorber. It can be seen from Figure 9b that the imaginary part of the complex permittivity of graphene/nano-$Fe_3O_4$/PLA composite absorber is higher than graphene/PLA and nano-$Fe_3O_4$/PLA composite absorbers, because in the microwave frequency band, the inherent electric dipole orientation polarization and interfacial polarization have the characteristics of large damping and small restoring force, which becomes the main factor affecting "$\varepsilon$". In addition, since graphene itself is superior in conductivity to nano-$Fe_3O_4$, the imaginary part of complex permittivity of graphene/PLA composite absorber is higher than that of nano-$Fe_3O_4$/PLA composite absorber.

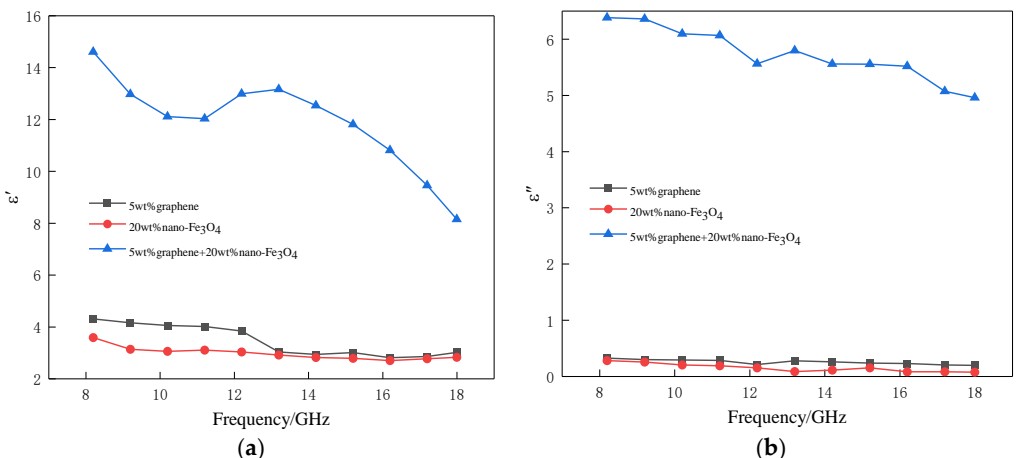

**Figure 9.** Complex permittivity of composite absorbers with different absorbing agent composition. (**a**) The real part of the complex permittivity; (**b**) the imaginary part of the complex permittivity.

The complex permeability of composite absorbers with different absorbing agent compositions are shown in Figure 10. Since the graphene/PLA composite absorber does not have magnetism, it does not participate in the comparison. As shown in Figure 10a, as the frequency increases, the real part of the complex permeability gradually decreases, because the magnetization speed of the nano-$Fe_3O_4$ in the composite absorber cannot keep up with the speed of the magnetic field change, resulting in the magnetization relaxation phenomenon [19]. The complex permeability of the graphene/nano-$Fe_3O_4$/PLA composite absorber is higher than that of the nano-$Fe_3O_4$/PLA composite absorber, because in addition to the magnetic effect of nano-$Fe_3O_4$, the graphene/nano-$Fe_3O_4$/PLA composite absorber has a synergistic effect between graphene and nano-$Fe_3O_4$ [20]. Therefore, the addition of graphene improves the magnetic storage capacity of the material. It can be seen from Figure 10b that as the frequency increases, the imaginary part of the complex permeability first rises and then falls, and a resonance peak appears at 16 GHz, which is mainly caused by the natural

resonance of the nano-$Fe_3O_4$ in the composite absorber. It is worth noting that the imaginary part of the composite permeability of the graphene/nano-$Fe_3O_4$/PLA composite absorber is higher than that of the nano-$Fe_3O_4$/PLA composite absorber, and the resonance peak is obviously improved. This is due to the fact that the addition of graphene leads to an increase in the anisotropy effect of the nano-$Fe_3O_4$ in the composite absorber, resulting in an increase in the magnetic loss of the electromagnetic wave.

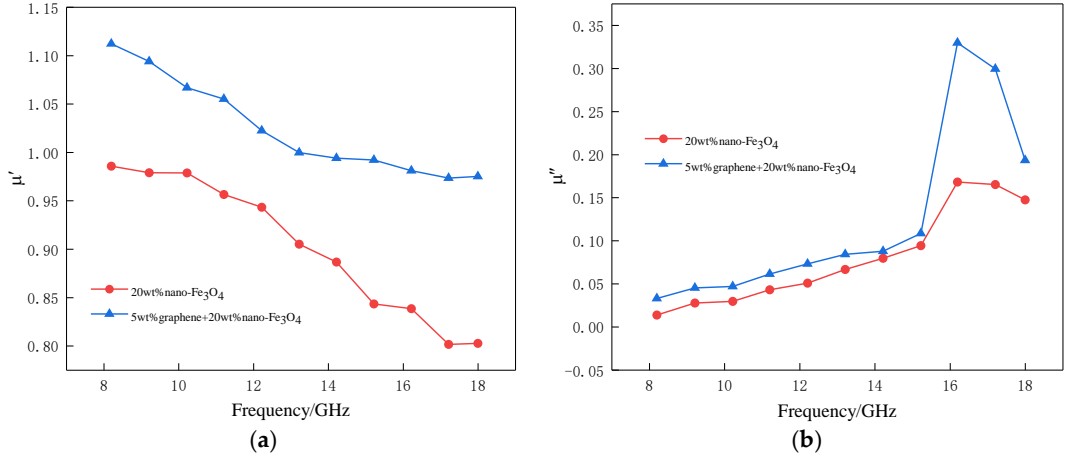

**Figure 10.** Complex permeability of composite absorbers with different absorbing agent composition. (**a**) The real part of the complex permeability; (**b**) the imaginary part of complex permeability.

The composite absorbing agent can realize the synergy of various loss mechanisms because it easily generates interfacial polarization [20]. The absorbing mechanism is further revealed by the SEM cross section. Figure 11 shows the SEM cross section of pure PLA and the composite absorbers with different absorbing agent composition. Figure 11a is a pure PLA absorber, the surface of which is relatively flat without holes. Figure 11b is a graphene/PLA composite absorber and it can be clearly observed that the graphene sheet is coated on the surface of the PLA substrate. Figure 11c is a nano-$Fe_3O_4$/PLA composite absorber and it can be clearly seen that the nano-$Fe_3O_4$ magnetic microspheres are adsorbed on the surface of the PLA substrate with holes. Figure 11d shows the graphene/nano-$Fe_3O_4$/PLA composite absorber and it can be clearly observed that the nano-$Fe_3O_4$ magnetic microspheres are adsorbed on the graphene sheets with a uniform distribution. This is because the graphene sheet has a large specific surface area and sufficient active adsorption point, which can effectively prevent the agglomeration of the nano-$Fe_3O_4$ magnetic microspheres and also contributes to the adsorption of nano-$Fe_3O_4$ magnetic microspheres on the graphene sheets. At the same time, the nano-$Fe_3O_4$ magnetic microspheres increase the distance between the graphene sheets and avoid the agglomeration of graphene. When the electromagnetic wave enters the inside of the graphene/nano-$Fe_3O_4$/PLA composite absorber, the "protrusion" formed by the nano-$Fe_3O_4$ magnetic microspheres and the graphene sheets have a reflection effect on the electromagnetic wave, which generates multiple reflections inside the material, resulting in the propagation path of electromagnetic waves becoming longer, increasing the loss of electromagnetic waves. This shows that the absorbing property of material can be effectively improved by compounding different loss types of absorbing agents.

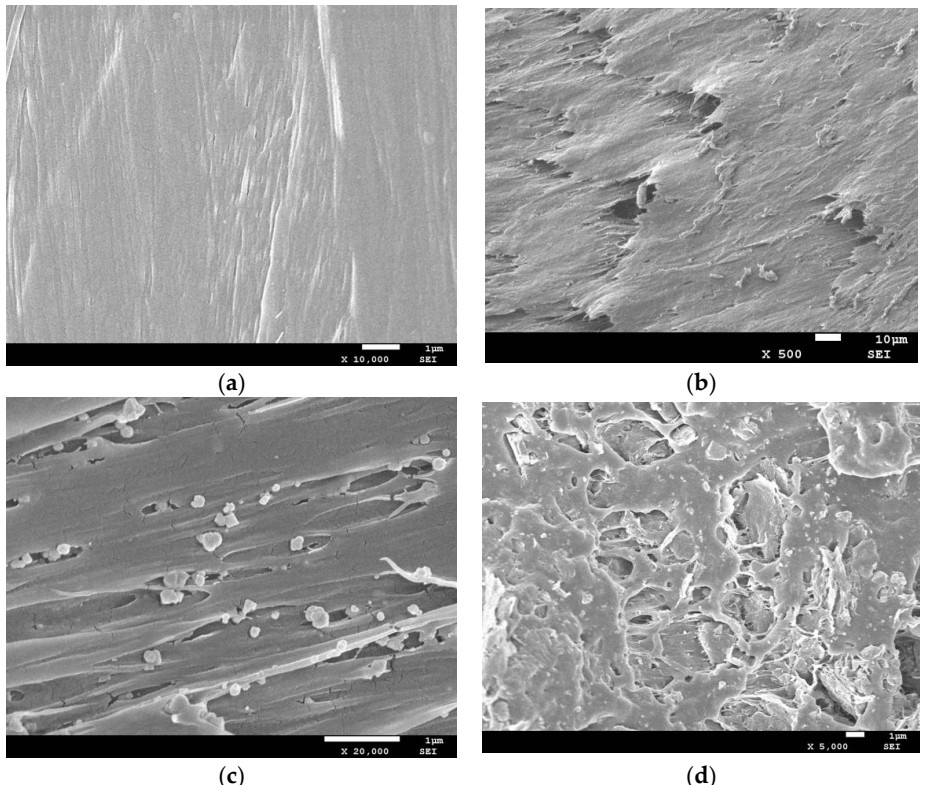

**Figure 11.** SEM cross-section of pure PLA and composite absorbers with different absorbing agent composition. (**a**) Pure PLA absorber; (**b**) 5 wt% graphene/PLA composite absorber; (**c**) 20 wt% nano-Fe$_3$O$_4$/PLA composite absorber; and (**d**) 20 wt% nano-Fe$_3$O$_4$ + 5 wt% graphene/PLA composite absorber.

## 4. Conclusions

The absorbing effect of three kinds of composite absorbers in the frequency range 8.2–18.0 GHz is compared and studied. In summary, we concluded that 7 wt% graphene/PLA composite absorber has better absorbing property than 5 wt% and 9 wt% graphene/PLA composite absorbers. Compared with 10 wt% and 30 wt% nano-Fe$_3$O$_4$/PLA composites, the 20 wt% nano-Fe$_3$O$_4$/PLA composite absorber has better absorbing property. The absorbing property of the graphene/nano-Fe$_3$O$_4$/PLA composite absorber consisting of 5 wt% graphene and 20 wt% nano-Fe$_3$O$_4$ are better than those of 5 wt% graphene/PLA composite absorber and 20 wt% nano-Fe$_3$O$_4$/PLA composite absorber. The absorbing mechanism is analyzed by analyzing the electromagnetic parameters and SEM. Due to the existence of the three-dimensional conductive network, the graphene/PLA composite absorbers absorb electromagnetic waves through the conductive and sheet structure. The nano-Fe$_3$O$_4$/PLA composite absorbers consume electromagnetic waves mainly through magnetic loss and the protrusions formed by the adsorption of nano-Fe$_3$O$_4$ magnetic microspheres on the surface of the substrate. Graphene/nano-Fe$_3$O$_4$/PLA composite absorbers combine the advantages of graphene/PLA and nano-Fe$_3$O$_4$/PLA composite absorbers. In addition, the synergistic effect of the multiple loss mechanism attenuated the electromagnetic waves effectively and achieved an efficient absorbing effect.

**Author Contributions:** Conceptualization, H.W.; L.X. wrote the paper. Y.C., L.L., E.H., B.L. and X.T. offered useful suggestions for the preparation. All authors have read and agreed to the published version of the manuscript.

**Funding:** This research was supported by the National Natural Science Foundation of China (grant no.51575313); the State Key Laboratory for Manufacturing Systems Engineering, Xi'an Jiaotong University (grant no. sklms2018001); and The Major Technological Innovation Project of Hubei Science and Technology Department (2019AAA164).

**Conflicts of Interest:** The authors declare no conflict of interest.

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
