# Peer review of "A Study on the Fused Deposition Modeling Process of Graphene/Nano-Fe3O4 Composite Absorber and its Absorbing Properties of Electromagnetic Microwave"

_applsci, doi:10.3390/app10041508_

Round 1

Reviewer 1 Report

The topic refers to absorption as the major phenomena in the study undertaken. The word absorption is a broad and generalised one and it is better to specify energy absorption or mater entity absorption in the topic. In fact the current study concentrates on the electromagnetic wave absorption by the composite structure. Introduction part may be extended in terms of reasoning the material selection for absorbent and the substrate and it can be correlated to possible applications. In the experimental method part, reasoning may be given for choosing the specified weight percentages, the sample size, printing temperature, printing speed and printing filling percentage. In the result and discussion part, a drastic change is observed in the complex permeability of imaginary part at a frequency of 16 GHz. There is no relevant information to back up the same in real part graph. This can be elaborated. Also the permittivity of the imaginary part does not show drastic variations at the aforementioned frequency. This phenomena may be elaborated. It necessitates that the study may be extended to higher frequencies since there exists a drastic change in energy absorbance at 16 GHz according to the imaginary part. The research paper exhibits high degree of novel research and the same may be accepted after the revisions recommended

Reviewer 2 Report

I attach the file

Round 2

Reviewer 2 Report

Thank you very much for the response